# Activated THP-1 Macrophage-Derived Factors Increase the Cytokine, Fractalkine, and EGF Secretions, the Invasion-Related MMP Production, and Antioxidant Activity of HEC-1A Endometrium Cells

**DOI:** 10.3390/ijms25179624

**Published:** 2024-09-05

**Authors:** Edina Pandur, Ramóna Pap, Katalin Sipos

**Affiliations:** 1Department of Pharmaceutical Biology, Faculty of Pharmacy, University of Pécs, 7624 Pécs, Hungary; pap.ramona@pte.hu (R.P.); katalin.sipos@aok.pte.hu (K.S.); 2National Laboratory of Human Reproduction, University of Pécs, 7624 Pécs, Hungary

**Keywords:** endometrium, macrophage, receptivity, cytokines, oxidative stress, fractalkine

## Abstract

Endometrium receptivity is a multifactor-regulated process involving progesterone receptor-regulated signaling, cytokines and chemokines, and additional growth regulatory factors. In the female reproductive system, macrophages have distinct roles in the regulation of receptivity, embryo implantation, immune tolerance, and angiogenesis or oxidative stress. In the present study, we investigated the effects of PMA-activated THP-1 macrophages on the receptivity-related genes, cytokines and chemokines, growth regulators, and oxidative stress-related molecules of HEC-1A endometrium cells. We established a non-contact co-culture in which the culture medium of the PMA-activated macrophages exhibiting the pro-inflammatory phenotype was used for the treatment of the endometrial cells. In the endometrium cells, the expression of the growth-related factors activin and bone morphogenetic protein 2, the growth hormone EGF, and the activation of the downstream signaling molecules pERK1/2 and pAkt were analyzed by ELISA and Western blot. The secretions of cytokines and chemokines, which are involved in the establishment of endometrial receptivity, and the expression of matrix metalloproteinases implicated in invasion were also determined. Based on the results, the PMA-activated THP-1 macrophages exhibiting a pro-inflammatory phenotype may play a role in the regulation of HEC-1A endometrium cells. They alter the secretion of cytokines and chemokines, as well as the protein level of MMPs of HEC-1A cells. Moreover, activated THP-1 macrophages may elevate oxidative stress protection of HEC-1A endometrium cells. All these suggest that pro-inflammatory macrophages have a special role in the regulation of receptivity-related and implantation-related factors of HEC-1A cells.

## 1. Introduction

Macrophages are crucial immune cells of the innate immune system [1]. They have specific functions in the protection against infections [2,3]. They help in the activation of lymphocytes and play a part in the anti-cancer system [4,5]. Macrophages are present in the endometrium throughout the female cycle, and they regulate tissue breakdown, repair, and tissue remodeling during pregnancy [6,7]. Also, they have a unique role in the development of endometrial receptivity and embryo implantation [8]. Macrophages contribute to the indirect regulation of both processes by secreted factors, e.g., growth factors, pro-inflammatory cytokines, chemokines, and other molecules [9]. 

Activation of monocytes can result in the differentiation into different types of macrophages with various functions. These functions include phagocytosis, antigen presentation, cytokine and chemokine production, or cytotoxicity [10]. The major types of macrophages [10], M1 and M2, have distinct roles in the inflammatory process. M1 macrophages are involved in pro-inflammatory responses, while M2 macrophages are implicated in anti-inflammatory responses [11].

In the female reproductive system, macrophages play a crucial role in various stages of pregnancy, including endometrial receptivity, embryo implantation, and placental development [12]. During the implantation window, M1 macrophages are initially abundant in the endometrium and contribute to the inflammatory response required for embryo attachment and invasion [13]. However, with embryo implantation and the onset of placental development, a shift towards a more M2 phenotype is observed. M2 macrophages are essential for the development of immune tolerance, promotion of tissue remodeling, and support of placental angiogenesis and vasculogenesis [13,14,15]. M2 macrophages can be divided into M2a, M2b, M2c, and M2d subgroups. These macrophages differ in their cell surface markers, secreted cytokines, and biological functions [16,17]. The dynamic interaction between M1 and M2 macrophages and other immune cell types is crucial for successful pregnancy. Imbalances in macrophage phenotypes have been implicated in various pregnancy complications, such as recurrent miscarriages and pre-eclampsia [13,14,18].

Endometrium receptivity is a complex procedure involving hundreds of interconnected, cooperating molecules and processes [19]. These molecules include progesterone receptors, regulating transcription factors like AP-1, JUN, FOS, GATA2 [20], and SOX-17 [21], cell signaling regulators implicated in cell growth and proliferation (e.g., activin, follistatin, bone morphogenetic proteins) [22], cytokines, and chemokines such as IL-6, leukocyte inhibitory factor (LIF), TGFβ, IL-8, and fractalkine influencing the inflammatory processes [23,24,25,26,27]. Other factors like matrix metalloproteinases (MMP) and tissue inhibitors of matrix metalloproteinases (TIMP) strictly control invasion and embryo implantation [28]. 

The controlled production of reactive oxygen species and the proper balance between pro-oxidants and antioxidants have an essential role in female reproduction, including receptivity [29,30]. The abnormal increase in ROS production can lead to polycystic ovary syndrome, endometriosis, and pre-eclampsia [31].

In the present study, we investigated the effects of PMA-activated THP-1 macrophages on the receptivity-related genes, cytokines and chemokines, growth regulators, and oxidative stress-related molecules of HEC-1A endometrium cells. We established a non-contact co-culture in which the medium of the PMA-activated THP-1 cells exhibited the pro-inflammatory phenotype [32] was used for the treatment of HEC-1A endometrial cells. 

It has been revealed that activated THP-1 macrophage-derived factors enhanced the expression of growth-related factors activin and BMP2, as well as the growth hormone EGF. The activation of the downstream signaling molecules pERK1/2 and pAkt and the activation of cyclin D expression were observed by Western blot and real-time PCR analysis in this in vitro system. Also, they triggered the secretion of fractalkine, IL-6, LIF, and TGFβ of HEC-1A endometrial cells. Activated THP-1 macrophage-derived molecules induced the production of MMP2 and MMP9 and reduced the protein expression of MMP inhibitors, TIMP1, and TIMP2 of HEC-1A cells. Finally, the total antioxidant capacity, as well as catalase and superoxide dismutase 2 enzyme activities, were induced in HEC-1A endometrium cells. 

Based on the results, we suppose that activated THP-1 macrophages with pro-inflammatory phenotype may play a role in the regulation of cytokine and chemokine secretions of HEC-1A cells as well as the expression of MMPs, which are implicated in endometrium receptivity and invasion. Moreover, activated THP-1 macrophages may contribute to increased oxidative stress protection. In summary, activated THP-1 macrophages with a pro-inflammatory phenotype might be involved in the regulation of endometrial receptivity-related and implantation-related factors of HEC-1A cells.

## 2. Results

### 2.1. Macrophage-Derived Factors Regulate the Expression of Receptivity-Related Genes of HEC-1A Cells

After the treatments of HEC-1A cells with the conditioned medium of the activated THP-1 cells, the relative mRNA expression levels of endometrium receptivity-related genes were determined by real-time PCR. The macrophage-derived factors significantly induced the expression of activin in the 24 h and 48 h experiments (Figure 1A). Meanwhile, follistatin (the activin inhibitor) mRNA levels significantly decreased using the 24 h conditioned medium but significantly increased using the 48 h conditioned THP-1 cell culture medium (Figure 1A). 

BMP2 mRNA levels were slightly elevated in HEC-1A cells after the treatments. The BMP2 protein measurements showed a significant rise both in the 24 h and 48 h conditioned, PMA-activated THP-1 medium (PMAi) (Figure 1B). After the treatments of HEC-1A cells with the conditioned THP-1 medium, the secreted BMP2 protein levels significantly lowered (PMA) compared to the initial BMP2 concentrations (PMAi) measured in the conditioned, PMA-activated THP-1 medium (Figure 1B), suggesting the interaction of BMP2 with its receptor and the activation of the cell signaling pathways. 

No considerable alterations were observed in PR, SRC-1 co-receptor, and SOX-17 mRNA expression levels (Figure 1C). At the protein level, PR, phospho-PR, SRC-1, and SOX-17 levels were significantly decreased in HEC-1A cells after incubation with the 24 h or 48 h conditioned, PMA-activated THP-1 medium (Figure 1D–H). 

Based on the results, it can be supposed that the macrophage-derived factors act on the activin receptor and BMPR signaling pathways by modifying the ligand expression, which can influence the cell cycle of the endometrial cells. On the other hand, they decrease the levels of the receptivity-related proteins.

### 2.2. Macrophage-Derived Factors Modify the Expression of Fractalkine and Its Receptor CX3CR1 of HEC-1A Cells

The fractalkine/CX3CR1 axis is an important regulator of endometrial receptivity and implantation. The FKN mRNA expression also exhibited an augmenting pattern with significant elevation after the treatment using the 48 h conditioned THP-1 culture medium (Figure 1A), and the CX3CR1 mRNA levels significantly increased in both experiments (Figure 2A). 

The FKN ELISA measurement demonstrated an elevation in the THP-1 medium after the activation (PMAi), which showed a significant increment in the case of HEC-1A cells (PMA) (Figure 2B), suggesting that the activated macrophages contribute to the further FKN secretion and maybe the activation of the FKN/CX3CR1 axis in HEC-1A cells.

At protein level, CX3CR1 was significantly raised in HEC-1A cells compared to the control HEC-1A cells when the cells were cultured in the 24 h-conditioned THP-1 culture medium but did not change after the 48 h-conditioned medium treatment (Figure 2C,D). 

### 2.3. The Effect of Macrophage-Derived Factors on the Cytokine Production of the Endometrium Cells

The levels of secreted cytokines related to endometrial receptivity were also monitored. The induction of THP-1 cells caused almost no alteration in the level of LIF (Figure 3A) but significantly increased the IL-6, IL-8, and TGFβ protein levels, showing the activation of THP-1 cells (PMAi) (Figure 3B–D). These initial cytokine concentrations were significantly elevated by HEC-1A cells in the case of LIF, IL-6, and TGFβ (PMA) (Figure 3A,B,D). Still, a minor decrease was observed in the case of IL-8 secretions (Figure 3C) after the treatment of HEC-1A cells with the conditioned culture medium. The alterations in the cytokine secretions of the endometrium cells showed an interrelationship with the time of the incubation of the THP-1 culture medium (24 h or 48 h).

### 2.4. Macrophage-Related Factors Alter the EGF Secretion, Akt, and ERK1/2 Activation, Regulating the Growth and Proliferation of the Endometrium Cells

Since an elevation in the expression of activin, which can control the growth of the endometrium cells, was found in HEC-1A cells after conditioned medium treatment, the EGF signaling pathway, another regulator of cell proliferation, was also examined. The EGF protein levels were increased in the PMA-activated THP-1 cell culture medium, and a significant further elevation was observed after treating HEC-1A cells with the conditioned medium, suggesting that the macrophage-derived factors triggered the growth hormone production of the endometrial cells (Figure 4A). 

The activation of the downstream signaling molecules was also investigated. The phosphorylated Akt levels were significantly raised in HEC-1A endometrial cells in both experiments, although the 24 h conditioned medium was more effective on that (Figure 4B,D). In the case of pERK1/2, the phosphorylated protein levels showed a significant augmentation after the 24 h conditioned THP-1 medium treatment but a nonsignificant reduction after using the 48 h conditioned medium (Figure 4B,C). The mRNA expression of cyclin D, a cell cycle regulator, and an ERK1/2 target gene, significantly increased, suggesting the alteration of the cell cycle regulation (Figure 4E). The resazurin-based cell viability measurement revealed significantly elevated percentages compared to the controls (Figure 4E). These results were supported by the determination of ATP concentration, which also showed a significant rise (Figure 4F). The increasing ATP levels indicate a higher living cell number.

### 2.5. Macrophage-Related Factors Alter the MMP2, and MMP9 Secretion and TIMP1 and 2 Involved in Invasion

Together, the MMPs and TIMP proteins regulate the invasion and can help in embryo implantation. We examined the evolution of protein secretion and expression in endometrial cells after treatments with the conditioned THP-1 media. The MMP2 concentration was elevated after the activation of the THP-1 cells. Still, its level was further significantly augmented after the treatments of HEC-1A cells (Figure 5A). In the case of MMP9, a smaller but still significant increase was revealed (Figure 5B). 

In parallel with the protease secretion, the inhibitor TIMP1 protein levels were significantly lowered (Figure 5C,D). In the case of TIMP2, only the 48 h conditioned THP-1 culture medium treatment caused a significant downregulation at the protein level (Figure 5B,D). 

These alterations in the MMP and TIMP protein levels in HEC-1A cells may help in invasion and may play a role during implantation.

### 2.6. Macrophages Trigger the Antioxidant Protection of the Endometrium Cells

Macrophages can contribute to oxidative stress by generating reactive oxygen species, which are involved in the protection against infections. It was investigated whether the endometrial cells can defend themselves from this harmful condition. 

The protein levels of the Nrf2 transcription factor regulating the expression of antioxidant enzymes SOD2 and catalase were analyzed in HEC-1A cells treated with the conditioned, PMA-activated THP-1 culture medium. The Nrf2 protein levels were significantly elevated in HEC-1A cells (Figure 6A,B), suggesting an increase in ROS production. The Western blot analyses of the antioxidant enzymes, catalase, and SOD2 also revealed a strong and significant upregulation at the protein level (Figure 6A,C,D).

The total antioxidant capacity of the treated HEC-1A endometrial cells showed significant elevation in both types of experiments, suggesting that the macrophage-derived factors may provoke a defence against oxidative stress (Figure 6E). 

## 3. Discussion

The receptive endometrium is essential for embryo implantation. Endometrial receptivity is regulated by several factors, which act together at the right time and place for preparing the inner lining of the uterus for pregnancy [19,20,24]. The endometrium is a complex, dynamic tissue that changes throughout the female cycle. At the proliferative phase, the endometrial cells divide and increase the thickness of the tissue. During the secretory phase, the endometrium releases cytokines, growth factors, and adhesion molecules for successful implantation. These regulatory factors are released by the cells of the immune system (NK cells, dendritic cells, and macrophages) as well as the endometrium cells themselves [33]. According to the involvement of the immune system in the development of endometrium receptivity and embryo implantation, they are described as pro-inflammatory events [34,35].

Macrophages have a pivotal role in the establishment of endometrial receptivity, regulating the breakdown, repair, and remodeling of the endometrium [7,8]. It has been described that different populations of macrophages are present in the endometrium: the tissue-resident macrophages, which exhibit the M2 phenotype and possess high phagocytic activity, and the bone marrow-originated macrophages that continuously migrate into the uterus and can differentiate into M1 or M2 phenotypes [13]. The macrophage phenotype is influenced by a complex interplay of factors, including the local microenvironment, signaling molecules, and the specific stimulus [36]. M1 macrophages are responsible for pro-inflammatory responses, while M2 macrophages are involved in anti-inflammatory responses [11,17]. All types of macrophages originate from M0, which is a resting state of macrophages without a specific function [37].

The M1-type macrophages are bountiful in the receptive endometrium [1,8] and indispensable for implantation. They release growth factors, pro-inflammatory cytokines, chemokines, and other molecules [9] that can modify the expression of receptivity-related genes and proteins of the endometrium cells. There are different subgroups of M2-type macrophages. M2a macrophages are mainly activated by the cytokines IL-4 and IL-13. They become dominant at a later stage of implantation and play an important role in the development of immune tolerance as they are able to reduce the inflammatory response and promote tissue regeneration. M2b macrophages are activated by stimulation with immune complexes or peptides of pathogenic origin [38]. Their role during implantation is less well understood, but they may contribute to feto-maternal tolerance and placental development. M2c macrophages are formed in response to IL-10 and have a predominantly immunosuppressive function. These cells may promote feto-maternal tolerance during implantation and prevent fetal rejection. M2d macrophages are formed under nutrient-deficient conditions and are primarily involved in tissue regeneration and energy storage. Thus, they may be important for placental development during implantation [16,17].

In the present study, a PMA-activated-THP-1 macrophage/HEC-1A endometrium non-contact co-culture was established, in which only the conditioned medium of the macrophages was used to treat HEC-1A endometrial cells. The PMA-mediated differentiation of the THP-1 cells directs the macrophages into the pro-inflammatory state [32]. We focused on the genes and proteins that have significant functions in the evolution of endometrial receptivity. 

The progesterone receptor signaling is crucial in the development of receptive endometrium [20]. At the proliferative phase, the PR expression increases, while during the secretory phase, it decreases. The downregulation of PR signaling is essential for successful implantation [39]. The PR expression, as well as its co-receptor SRC-1 and the PR-regulated downstream SOX-17 transcription factor [25] mRNA expression levels, were significantly reduced in the endometrial cells after the treatments with the conditioned macrophage medium. These alterations suggest the downregulation of PR signaling pathways, such as the NFκB, STAT3, and AP-1 pathways [40]. On the other hand, the proliferation regulators activin and BMP2, acting on distinct cell surface receptors, the activin receptor, and the BMP receptor [22] showed significant elevation, suggesting the role of pro-inflammatory macrophages in the cell cycle regulation of endometrium cells. In our previous study, it has been revealed that activin and BMP2 function as growth factors [27].

The pro-inflammatory macrophages secrete numerous inflammatory cytokines and chemokines regulating endometrium receptivity [24,26]. The treatment of HEC-1A cells with macrophage-derived factors containing culture medium triggered the secretion of leukocyte inhibitory factor (LIF), interleukin-6 (IL-6), and transforming growth factor β (TGFβ), showing that the macrophages themselves contribute to endometrium receptivity by cytokine production. Moreover, they further induce the cytokine synthesis of the endometrium cells, proving both the direct and indirect effects of macrophages on the endometrium cells. In the case of IL-8, no significant alteration was revealed, which can be explained by the fact that IL-8 stimulates trophoblast migration and invasion for embryo implantation [41]. On the other hand, the high level of IL-8 contributes to the hyperproliferation of the endometrium cells, which can lead to endometriosis [42,43].

Fractalkine has been described as the regulator of endometrium receptivity, embryo implantation, and maternal–fetal communication [25,27,44]. The expression of the cytokines of HEC-1A endometrial cells may be prompted by the fractalkine/fractalkine receptor (CX3CR1) axis via the NFκB signaling pathway [45]. The FKN secretion from the treated endometrium cells was elevated, and the protein level of its receptor was upregulated, suggesting the increased activation of this signaling pathway contributing to cytokine transcription and cell proliferation by inhibiting apoptosis [46]. 

The epidermal growth factor (EGF) is essential in mediating the proliferation of endometrial cells during the proliferative phase to prepare the endometrium for implantation [47,48]. Based on the results, the pro-inflammatory macrophages release EGF, and they enhance the EGF secretion of the endometrium cells and strengthen the activation of ERK1/2 and Akt serine/threonine protein kinases, which are involved in the survival, proliferation, and growth of the cells [49,50] in which the activation of cyclin D expression has a pivotal role [51]. The same transcription factors are involved in FKN signaling, suggesting cooperation between the two distinct pathways in the regulation of apoptosis and cell division [52,53].

According to the effects of macrophages on the endometrium cells, the interconnected signaling pathways regulated by cytokines, FKN, and EGF are critical in the development of endometrial receptivity. LIF belongs to the IL-6 cytokine family and regulates the mitogen-activated protein kinase (MAPK), the Janus kinase (Jak)/STAT, or phosphatidylinositol 3-kinase (PI3K) signaling pathways via its receptor, LIFR [54]. On the other hand, the expression of LIF is controlled by TGFβ, activin-A, and IL-6, showing a crosstalk between the cytokine signaling and the complex regulation of endometrium receptivity [55,56] by macrophages and the endometrium itself.

The matrix metalloproteinases (MMPs) and their inhibitors, the tissue inhibitors of matrix metalloproteinases (TIMP), are involved in the precise control of the invasion and embryo implantation [28] and also in inflammation [57]. The pro-inflammatory macrophages release MMPs at relatively high concentrations, which may contribute to the invasion of the trophoblast cells into the endometrium. The substances they secrete increase endometrial cell MMP production and decrease inhibitor secretion, which may further aid the processes of invasion and implantation.

IL-6, LIF, and FKN secreted by both macrophages and endometrial cells drive MMP2 and MMP9 expression and synthesis via the MAPK signaling pathway, demonstrating a convergence between the different cell surface receptors [53,58,59].

Inflammatory processes can induce the production of reactive oxygen species (ROS) and can lead to oxidative stress, which can be harmful to the endometrium and receptivity [29,30]. The pro-inflammatory macrophages also release ROS, even at high concentrations, to eliminate infections [60]. The overproduction of ROS by macrophages can result in the development of endometriosis [61]. Therefore, balanced ROS production and protection against them are required for the proper function of the endometrium. After incubation with the conditioned macrophage medium, the HEC-1A cells significantly increase their Nrf2 transcription factor level, which in turn can activate the expression of the antioxidant enzymes SOD2 and catalase, leading to an elevated total antioxidant capacity and providing an antioxidant defense [62,63]. A summary of the results and the signaling pathways that are implicated in the proposed mechanism of action can be found in Figure 7.

The study revealed the regulatory function of pro-inflammatory THP-1 macrophages in the induction of IL-6, LIF, and TGFβ cytokines and FKN chemokine of HEC-1A endometrial cells, which are pivotal in the evolution of receptivity. Furthermore, the PMA-activated THP-1 macrophage-derived molecules enhanced the EGF signaling and increased cyclin D mRNA expression, which may be influenced by the cytokine and FKN signaling pathways as well as activin and BMP2 signal transduction, revealing a complex and multifactorial network acting in the management of receptivity-related proteins. The PMA-activated THP-1 macrophage-derived molecules induced the secretion of MMP2 and MMP9 and reduced the protein expression of TIMP1 and 2, suggesting the role of pro-inflammatory macrophages in the regulation of invasion-related proteins. Finally, the total antioxidant capacity, as well as the catalase and superoxide dismutase 2 enzyme activities, were induced in HEC-1A endometrium cells and may elevate oxidative stress protection. 

The investigation provides a deeper insight into the regulation of the endometrium by pro-inflammatory macrophages. However, the study has some limitations, since an in vitro system with PMA-activated THP-1 cells and their culture medium and HEC-1A endometrial cells were used in experiments that prevented the direct interaction between the two cell types. In the non-contact co-culture, only the effects of the secreted molecules can be considered on the HEC-1A cells. Moreover, the endometrium is a heterogeneous tissue containing different cell types like epithelial, glandular, stromal, vascular, and immune cells [64]. The HEC-1A cell line represents the phenotype of luminal or glandular epithelial cells [65]. In the regulation of the endometrium, the fine-tuning and balance of the different types of macrophages are necessary. Therefore, we plan to analyze the effects of M2-type macrophages on the endometrium receptivity. Moreover, a suitable in vivo experimental model would be beneficial in describing the roles of macrophage-derived factors in the development of endometrium receptivity and, later, the invasion.

## 4. Materials and Methods

### 4.1. Cell Cultures and Treatments

The THP-1 human monocyte/macrophage cell line was purchased from Merck (Merck Life Science Kft., Budapest, Hungary). The THP-1 cells were cultured in RPMI-1640 culture medium (Capricorn Scientific GmbH, Ebsdorfergrund, Germany) supplemented with 10% fetal bovine serum (FBS; Capricorn Scientific GmbH, Ebsdorfergrund, Germany) and 1% penicillin/streptomycin (P/S; Capricorn Scientific GmbH, Ebsdorfergrund, Germany). Phorbol 12-myristate 13-acetate (PMA) was purchased from Merck (Merck Life Science Kft., Budapest, Hungary). A 1 mg/mL PMA stock solution was used for the activation of the THP-1 cells in the final concentration of 100 ng/mL for 24 h treatment [32]. After the PMA treatment, the culture medium was removed, and the activated macrophages were washed with 1× phosphate buffer saline (PBS; Capricorn Scientific GmbH, Ebsdorfergrund, Germany). Then, THP-1 cells were incubated in fresh, complete RPMI-1640 medium for 24 h or 48 h. After the incubation periods, the conditioned medium was transferred to HEC-1A endometrial cells for 24 h. The HEC-1A human endometrial cells were purchased from ATCC (ATCC HTB-112). The cells were maintained in McCoy’s 5A medium with Ishikawa and Grace modification (Corning Inc., Corning, NY, USA). They were supplemented with 10% fetal bovine serum (FBS, Capricorn Scientific GmbH, Ebsdorfergrund, Germany) and 1% penicillin/streptomycin (P/S, Capricorn Scientific GmbH, Ebsdorfergrund, Germany). The endometrium cell cultures were separated into four groups: (1) HEC-1A cells treated with 24 h non-activated THP-1 medium; (2) HEC-1A cells treated with 24 h conditioned, PMA-activated THP-1 medium; (3) HEC-1A cells treated with 48 h non-activated THP-1 medium; (4) HEC-1A cells treated with 24 h conditioned, PMA-activated THP-1 medium. The first and the third groups were the controls, in which the medium was also incubated with non-activated THP-1 cells for 24 h or 48 h. The cells were cultured in a humified atmosphere in the presence of 5% CO_2_ at 37 °C.

### 4.2. Real-Time PCR

For the mRNA expression analysis, THP-1 cells were cultured in 6-well plates (Biologix Europe, Hallbergmoos, Germany) using 5 × 10^5^ cells/well and were treated with PMA as described above. HEC-1A endometrial cells were placed onto 6-well plates using 5 × 10^5^ cells/well and were treated with the non-activated and conditioned, PMA-activated THP-1 medium, as defined in Section 2.1. After incubation, HEC-1A cells were collected by trypsinization and centrifugation, and then the cell pellets were washed with 1× PBS. Total RNA was isolated with an Aurum Total RNA Mini Kit (Bio-Rad Inc., Hercules, CA, USA). The cDNA synthesis was performed using the iScript cDNA Synthesis Kit (Bio-Rad Inc., Hercules, CA, USA). The real-time PCR reactions were carried out in a CFX96 Opus Real-Time PCR System (Bio-Rad Inc., Hercules, CA, USA) using reagent iTaq Universal SYBR Green Reagent Mix (Bio-Rad Inc., Hercules, CA, USA). Each reaction tube contained 7.2 μL of water, 10 μL of 2× Master Mix, 10 μM of forward and reverse primers, and 20 ng of cDNA in a total reaction volume of 20 µL. The cycles were as follows: 95 °C for 5 min first denaturation; 95 °C for 5 s, 56 °C for 30 s, 72 °C for 30 s repeated for 45 cycles; 72 °C for 5 min final extension. The specificity of the reactions was proven by generating a melting curve after each run (60 °C–95 °C with 0.5 s/°C ramp rate). The relative mRNA expression levels (fold change) of the target genes were calculated by the Livak (∆∆Ct) method using the Bio-Rad CFX Maestro 2.3. software (Bio-Rad Inc., Hercules, CA, USA). Glyceraldehyde 3-phosphate dehydrogenase (GAPDH) was used as a normalization gene for real-time PCR reactions [66]. The mRNA expression levels of the target genes in the control cells were regarded as 1. The measurements were performed in triplicate in three independent experiments. The primer sequences used in the experiments are presented in Table 1. 

### 4.3. Western Blot

For the Western blot experiments, both cell types were placed onto 6 cm wide cell culture dishes (Biologix Europe, Hallbergmoos, Germany) using 10^6^ cells/dish. The endometrial cell cultures were treated with the conditioned THP-1 medium as described earlier. After the treatments, the cells were collected by centrifugation and were lysed in 200 µL of ice-cold lysis buffer (50 mM Tris-HCl, 150 mM NaCl, 0.5% Triton X-100, pH 7.4) and were supplemented with Complete Mini protease inhibitor cocktail (Roche Ltd., Basel, Switzerland). The protein content of the samples was determined using the DC Protein Assay Kit (Bio-Rad Laboratories, Hercules, CA, USA) and the MultiSkan GO spectrophotometer (Thermo Fisher Scientific Inc., Waltham, MA, USA) at 750 nm. Identical amounts of proteins from each sample were separated in SDS-polyacrylamide gels using the Mini Protean Tetra Cell equipment (Bio-Rad Laboratories, Hercules, CA, USA). Then, the proteins were blotted onto nitrocellulose membranes (Amersham Biosciences, GE Healthcare, Amersham, UK). 

The membranes were blocked with a TBST buffer containing 5% (*w*/*v*) non-fat dry milk at room temperature for 1 h. The following primary antibodies were used in the Western blot according to the manufacturers’ protocols: anti-PR IgG (1:1000, 1 h, room temperature, Thermo Fisher Scientific Inc., Waltham, MA, USA); anti-P-PR IgG (1:1000, 1 h, room temperature, Thermo Fisher Scientific Inc., Waltham, MA, USA); anti-SRC-1 IgG (1:1000, 1 h, room temperature, Thermo Fisher Scientific Inc., Waltham, MA, USA); anti-SOX-17 IgG (1:1000, 1 h, room temperature, Merck Life Science Kft., Budapest, Hungary); anti-CX3CR1 IgG (1:1000, 1 h, room temperature, Thermo Fisher Scientific Inc., Waltham, MA, USA); anti-pERK1/2 IgG (1:1000, overnight, 4 °C, Cell Signaling Technology Europe, Leiden, The Netherlands); anti-pAkt IgG (1:1000, overnight, 4 °C, Cell Signaling Technology Europe, Leiden, The Netherlands); anti-TIMP1 IgG (1:1000, 1 h, room temperature, Thermo Fisher Scientific Inc., Waltham, MA, USA); anti-TIMP2 IgG (1:1000, 1 h, room temperature, Thermo Fisher Scientific Inc., Waltham, MA, USA); anti-Nrf2 IgG (1:1000, overnight, 4 °C, Cell Signaling Technology Europe, Leiden, The Netherlands); anti-SOD2 IgG (1:1000, 1 h, room temperature, Thermo Fisher Scientific Inc., Waltham, MA, USA); and anti-catalase IgG (1:1000, 1 h, room temperature, Thermo Fisher Scientific Inc., Waltham, MA, USA). The glyceraldehyde 3-phosphate dehydrogenase (anti-GAPDH IgG, 1:3000; Merck Life Science Kft., Budapest, Hungary) was used as a loading control. The secondary antibody, goat anti-rabbit IgG HRP conjugated IgG, was used (1:3000; Merck Life Science Kft., Budapest, Hungary) for 1 h at room temperature [25,27].

The blots were visualized by the UVItec Alliance Q9 Advanced Imaging System (UVItec Cambridge Ltd., Cambridge, UK). The membranes were developed using the WesternBright ECL chemiluminescent substrate (Advansta Inc., San Jose, CA, USA). The protein levels were expressed as a percentage of the target protein/GAPDH ratio. Western blots shown in the figures were representative of three independent experiments. 

### 4.4. Enzyme-Linked Immunosorbent Assay (ELISA) Measurements

The supernatants of the untreated and PMA-treated THP-1 cells were collected and used to determine the initial concentration of the target protein secreted by the THP-1 cells. The supernatants of HEC-1A cells were collected and stored at −80 °C until the ELISA measurements. The secreted BMP2 concentration was measured by the Human BMP2 ELISA Kit (Merck Life Science Kft., Budapest, Hungary). The FKN concentrations were determined by the Human Fractalkine ELISA Kit (Thermo Fisher Scientific Inc., Waltham, MA, USA). The LIF and TGFβ concentrations were determined by using the Human LIF and Human TGFβ ELISA Kits (Merck Life Science Kft., Budapest, Hungary). The IL-6 and IL-8 secretions of the cells were measured by using the Human IL-6 and IL-8 ELISA Kits (Thermo Fisher Scientific Inc., Waltham, MA, USA). The MMP2 and MMP9 levels were determined by the Human MMP2 and MMP9 ELISA Kits (Merck Life Science Kft., Budapest, Hungary) [67]. The measurements were performed in triplicate in three independent experiments. All kits were used according to the manufacturers’ protocols. 

### 4.5. Cell Viability Measurement

The viability of HEC-1A cells was measured using the TOX8 cell viability assay (Merck KGaA, Darmstadt, Germany) after the treatments with the non-activated THP-1 culture medium (Ctrl) or PMA-activated THP-1 culture medium (PMA). Briefly, HEC-1A cells were seeded on a 96-well culture plate. After the treatments, 10 µL of resazurin reagent was added to each well, and the plate was incubated for 2 h at 37 °C and 5% CO_2_. The absorbance of HEC-1A cells was measured at 600 nm using a reference wavelength of 690 nm in a MultiSkan GO microplate spectrophotometer (Thermo Fisher Scientific Inc., Waltham, MA, USA) [68]. Viability was expressed as the percentile of the total cell number of the control cells cultured in a non-activated THP-1 medium.

### 4.6. ATP Concentration Determination

The ATP concentration of HEC-1A cells was measured with the ATP Determination Kit (Thermo Fisher Scientific Inc., Waltham, MA, USA) according to the manufacturer’s protocol. The kit provides a luminescence assay for the quantitative determination of ATP with recombinant firefly luciferase and its substrate D-luciferin. For the determination, 100 µL of reaction solution was added to 100 µL of cell samples [69]. The luminescence was measured by an EnSpire Multimode microplate reader (Perkin Elmer, Rodgau, Germany) in luminescence mode. The values were expressed as pM.

### 4.7. Total Antioxidant Capacity (TAC) Measurement

The THP-1 cells were cultured in 6-well plates (Biologix Europe, Hallbergmoos, Germany) using 5 × 10^5^ cells/well and were treated with PMA as described above. The HEC-1A endometrial cells were placed onto 6-well plates using 5 × 10^5^ cells/well and were treated with the non-activated and conditioned, PMA-activated THP-1 culture medium, as defined in Section 2.1. After the treatments, HEC-1A cells were collected, and the cells were lysed in ice-cold 1× PBS with sonication. Then, the samples were centrifuged at 14,000 rpm to pellet debris for 10 min. The TAC concentration was determined from 20 µL of the supernatants using the Total Antioxidant Capacity Assay Kit (Merck Life Science Kft., Budapest, Hungary). The results were expressed in µM Trolox equivalent [70].

### 4.8. Data Analysis

Statistical analysis was performed using SPSS software version 24.0 (IBM Corporation, Armonk, NY, USA). Statistical significance was determined by one-way ANOVA followed by Tukey’s post hoc test. Data are shown as the mean ± standard deviation (SD) and were considered statistically significant if the *p*-value was lower than 0.05. The experiments were repeated three times, and the number of technical replicates was three in the case of real-time PCR, ELISA, viability, ATP, and TAC measurements.

## Figures and Tables

**Figure 1 ijms-25-09624-f001:**
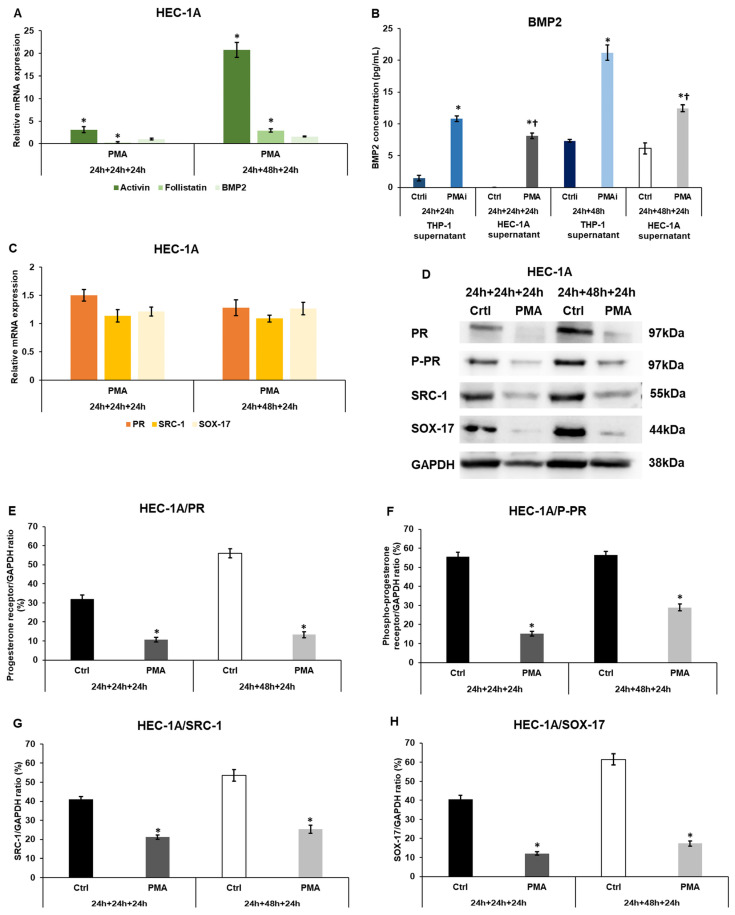
Real-time PCR analyses of activin, BMP2, follistatin, progesterone receptor, SRC-1 and SOX17, BMP2 ELISA, and Western blot analyses of the receptivity-related genes PR, P-PR, SRC-1, and SOX-17 in HEC-1A cells. (**A**) mRNA analysis of activin, follistatin, and BMP2 of HEC-1A cells. (**B**) Concentration determination of secreted BMP2 protein of the non-activated THP-1 cells (Ctrli), PMA-activated THP-1 cells (PMAi), non-activated THP-1 culture medium-treated HEC-1A cells (Ctrl), and PMA-activated conditioned medium-treated HEC-1A cells (PMA). (**C**) Relative mRNA expression analysis of PR, SRC-1, and SOX-17. (**D**) Western blot of PR, phospho-PR, SRC-1, and SOX-17 of HEC-1A cells. (**E**) Analysis of PR protein levels in HEC-1A cells. (**F**) Analysis of P-PR protein levels in HEC-1A cells. (**G**) Analysis of SRC-1 protein levels in HEC-1A cells. (**H**) Analysis of SOX-17 protein levels in HEC-1A cells. The real-time PCR was performed using a SYBR Green protocol and GAPDH as a housekeeping gene. The expression of the target genes was considered 1 in the control cells, which were treated with the non-activated THP-1-derived culture medium. The ELISA measurement was performed using the Human BMP2 ELISA Kit according to the manufacturer’s protocol. In the case of WB experiments, the same amount of protein from each sample was separated into SDS-polyacrylamide gels. After blotting, the primary antibodies were used according to the manufacturer’s protocols. GAPDH was used as the loading control. The blots are representative images. The samples in the Western blots are: 1. Ctrl 24 h + 24 h + 24 h: medium incubation on non-activated THP-1 cells for 24 h; medium incubation on non-activated THP-1 cells after medium exchange for 24 h; addition of 24 h conditioned medium of non-activated THP-1 cells to HEC-1A cells for 24 h. 2. PMA 24 h + 24 h + 24 h: medium incubation on PMA-activated THP-1 cells for 24 h; incubation on PMA-activated THP-1 cells after medium exchange for 24 h; addition of 24 h conditioned medium of PMA-activated THP-1 cells to HEC-1A cells for 24 h. 3. Ctrl 24 h + 48 h + 24 h: medium incubation on non-activated THP-1 cells 24 h; medium incubation on non-activated THP-1 cells after medium exchange for 48 h; addition of 48 h conditioned medium of non-activated THP-1 cells to HEC-1A cells for 24 h. 4. PMA 24 h + 24 h + 24 h: incubation on PMA-activated THP-1 cells for 24 h; incubation on PMA-activated THP-1 cells after medium exchange for 48 h; addition of 48 h conditioned medium of PMA-activated THP-1 cells to HEC-1A cells for 24 h. The columns show the mean values ± SD. In (**A**,**E**–**H**), the * means *p* < 0.05 compared to the control. In (**B**), the * shows *p* < 0.05 compared to Ctrli or Ctrl, and the † indicates *p* < 0.05 compared to PMAi. Abbreviations: BMP2, bone morphogenetic protein 2; PMA, phorbol 12-myristate 13-acetate; Ctrli, initial concentration of the protein in the control culture medium; PMAi, initial concentration of the protein in the culture medium of PMA-treated THP-1 cells; GAPDH, glyceraldehyde 3-phosphate dehydrogenase; PR, progesterone receptor, P-PR, phospho-progesterone receptor.

**Figure 2 ijms-25-09624-f002:**
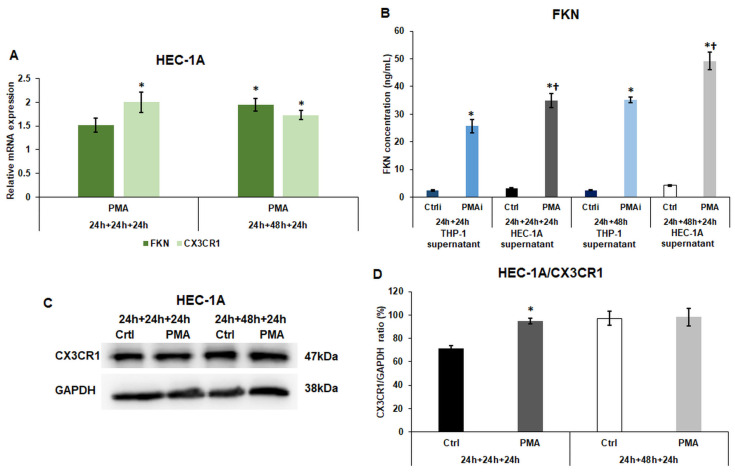
Real-time PCR analyses of FKN and CX3CR1, ELISA measurement of secreted FKN, Western blot analysis of CX3CR1. (**A**) Relative mRNA expression levels of FKN and CX3CR1 of HEC-1A cells. (**B**) Concentration determination of secreted FKN of the non-activated THP-1 cells (Ctrli), PMA-activated THP-1 cells (PMAi), non-activated THP-1 culture medium-treated HEC-1A cells (Ctrl), and PMA-activated conditioned medium-treated HEC-1A cells (PMA). (**C**) Western blot of CX3CR1 of HEC-1A cells. (**D**) Analysis of CX3CR1 protein levels in HEC-1A cells. The real-time PCR was performed using a SYBR Green protocol and GAPDH as a housekeeping gene. The expression of the target genes was considered 1 in the control cells, which were treated with the non-activated THP-1-derived culture medium. The ELISA measurement was carried out using the Human Fractalkine ELISA Kit according to the manufacturer’s protocol. In the case of WB experiments, the same amount of protein from each sample was separated into SDS-polyacrylamide gels. After blotting, the CX3CR1 antibody was used according to the manufacturer’s protocol. GAPDH was used as the loading control. The blots are representative images. The samples in the Western blots are: 1. Ctrl 24 h + 24 h + 24 h: medium incubation on non-activated THP-1 cells for 24 h; medium incubation on non-activated THP-1 cells after medium exchange for 24 h; addition of 24 h conditioned medium of non-activated THP-1 cells to HEC-1A cells for 24 h. 2. PMA 24 h + 24 h + 24 h: medium incubation on PMA-activated THP-1 cells for 24 h; incubation on PMA-activated THP-1 cells after medium exchange for 24 h; addition of 24 h conditioned medium of PMA-activated THP-1 cells to HEC-1A cells for 24 h. 3. Ctrl 24 h + 48 h + 24 h: medium incubation on non-activated THP-1 cells 24 h; medium incubation on non-activated THP-1 cells after medium exchange for 48 h; addition of 48 h conditioned medium of non-activated THP-1 cells to HEC-1A cells for 24 h. 4. PMA 24 h + 24 h + 24 h: incubation on PMA-activated THP-1 cells for 24 h; incubation on PMA-activated THP-1 cells after medium exchange for 48 h; addition of 48 h conditioned medium of PMA-activated THP-1 cells to HEC-1A cells for 24 h. The columns show the mean values ± SD. In (**A**,**D**), the * means *p* < 0.05 compared to the control. In (**B**), the * shows *p* < 0.05 compared to Ctrli or Ctrl, respectively, and the † indicates *p* < 0.05 compared to PMAi. Abbreviations: CX3CR1, fractalkine receptor; FKN, fractalkine; PMA, phorbol 12-myristate 13-acetate; Ctrli, initial concentration of the protein in the control culture medium; PMAi, initial concentration of the protein in the culture medium of PMA-treated THP-1 cells; GAPDH, glyceraldehyde 3-phosphate dehydrogenase.

**Figure 3 ijms-25-09624-f003:**
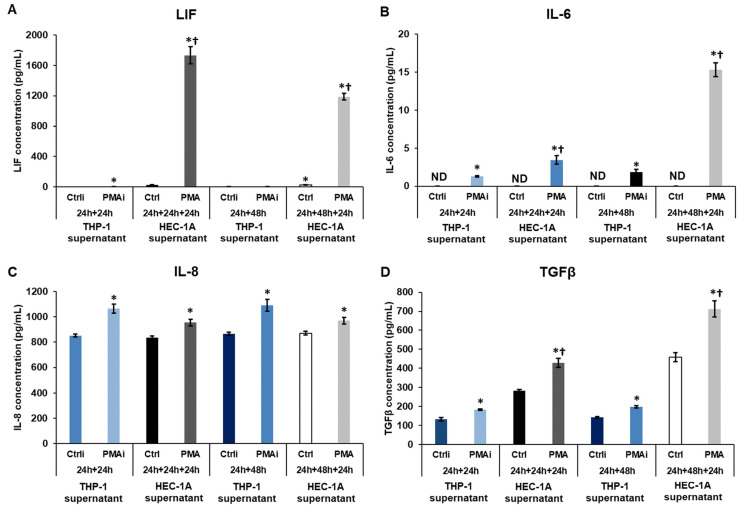
Concentration determinations of secreted LIF, IL-6, IL-8, and TGFβ of the treated endometrium cells. (**A**) Concentration determination of secreted LIF of the non-activated THP-1 cells (Ctrli), PMA-activated THP-1 cells (PMAi), non-activated THP-1 culture medium-treated HEC-1A cells (Ctrl), and PMA-activated conditioned medium-treated HEC-1A cells (PMA). (**B**) Concentration determination of secreted IL-6 of the non-activated THP-1 cells (Ctrli), PMA-activated THP-1 cells (PMAi), non-activated THP-1 culture medium-treated HEC-1A cells (Ctrl), and PMA-activated conditioned medium-treated HEC-1A cells (PMA). (**C**) Concentration determination of secreted IL-8 of the non-activated THP-1 cells (Ctrli), PMA-activated THP-1 cells (PMAi), non-activated THP-1 culture medium-treated HEC-1A cells (Ctrl), and PMA-activated conditioned medium-treated HEC-1A cells (PMA). (**D**) Concentration determination of secreted TGFβ of the non-activated THP-1 cells (Ctrli), PMA-activated THP-1 cells (PMAi), non-activated THP-1 culture medium-treated HEC-1A cells (Ctrl), and PMA-activated conditioned medium-treated HEC-1A cells (PMA). For the determination of secreted cytokine concentrations, human LIF, IL-6, IL-8, and TGFβ ELISA kits were used according to the instructions of the manufacturers. The columns show the mean values ± SD. The * means *p* < 0.05 compared to Ctrli or Ctrl, respectively, and the † indicates *p* < 0.05 compared to PMAi. Abbreviations: LIF, leukocyte inhibitory factor; IL-6, interleukin-6; IL-8, interleukin-8; TGFβ, transforming growth factor β; PMA, phorbol 12-myristate 13-acetate; Ctrli, initial concentration of the protein in the control culture medium; PMAi, initial concentration of the protein in the culture medium of PMA-treated THP-1 cells; ND, non-detected.

**Figure 4 ijms-25-09624-f004:**
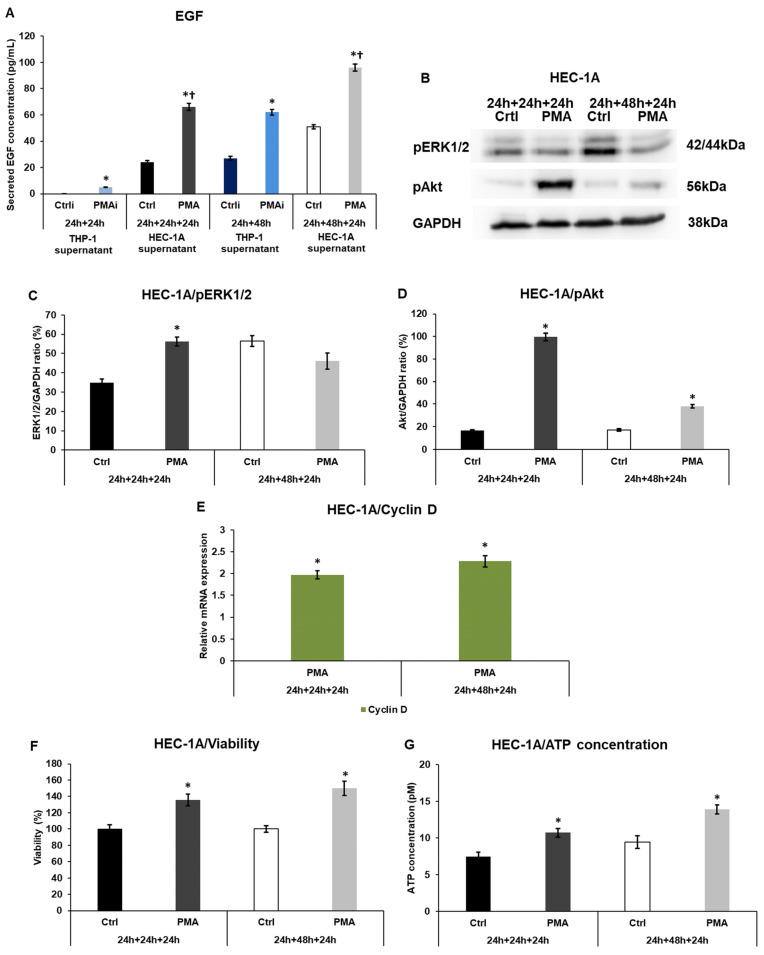
ELISA measurements of EGF, the Western blot analyses of phospho-Akt and phospho-ERK1/2 in HEC-1A cells, real-time PCR of cyclin D, and the viability and ATP levels of HEC-1A cells. (**A**) Concentration determination of secreted EGF of the non-activated THP-1 cells (Ctrli), PMA-activated THP-1 cells (PMAi), non-activated THP-1 culture medium-treated HEC-1A cells (Ctrl), and PMA-activated conditioned medium-treated HEC-1A cells (PMA). (**B**) Western blots of phospho-ERK1/2 and phospho-Akt of HEC-1A cells. (**C**) Analysis of pERK1/2 protein level of HEC-1A cells. (**D**) Analysis of pAkt protein levels of HEC-1A cells. (**E**) Relative mRNA expression analysis of cyclin D in HEC-1A cells. (**F**) Viability measurement of HEC-1A cells. (**G**) ATP concentration determination of HEC-1A cells. The samples in the Western blots, viability, and ATP measurements are: 1. Ctrl 24 h + 24 h + 24 h: medium incubation on non-activated THP-1 cells for 24 h; medium incubation on non-activated THP-1 cells after medium exchange for 24 h; addition of 24 h conditioned medium of non-activated THP-1 cells to HEC-1A cells for 24 h. 2. PMA 24 h + 24 h + 24 h: medium incubation on PMA-activated THP-1 cells for 24 h; incubation on PMA-activated THP-1 cells after medium exchange for 24 h; addition of 24 h conditioned medium of PMA-activated THP-1 cells to HEC-1A cells for 24 h. 3. Ctrl 24 h + 48 h + 24 h: medium incubation on non-activated THP-1 cells 24 h; medium incubation on non-activated THP-1 cells after medium exchange for 48 h; addition of 48 h conditioned medium of non-activated THP-1 cells to HEC-1A cells for 24 h. 4. PMA 24 h + 24 h + 24 h: incubation on PMA-activated THP-1 cells for 24 h; incubation on PMA-activated THP-1 cells after medium exchange for 48 h; addition of 48 h conditioned medium of PMA-activated THP-1 cells to HEC-1A cells for 24 h. A Human EGF ELISA kit was used to determine secreted EGF concentrations according to the manufacturer’s instructions. In the case of WB experiments, the same amount of protein from each sample was separated into SDS-polyacrylamide gels. After blotting, the phospho-Akt and phospho-ERK1/2 antibodies were used according to the manufacturer’s protocols. GAPDH was used as the loading control. The blots are representative images. The real-time PCR was performed using a SYBR Green protocol and GAPDH as a housekeeping gene. The expression of the target genes was considered 1 in the control cells, which were treated with the non-activated THP-1-derived culture medium. Cell viability was determined using the TOX8 Kit. The values were expressed as percentages compared to the controls. The viability of the controls was regarded as 100%. ATP determination was performed using the ATP Assay Kit, which was used according to the manufacturer’s protocol. The columns show the mean values ± SD. In (**A**), the * means *p* < 0.05 compared to Ctrli or Ctrl, respectively, and the † indicates *p* < 0.05 compared to PMAi. In (**C**–**G**), the * shows *p* < 0.05 compared to the control. Abbreviations: EGF, epidermal growth factor; PMA, phorbol 12-myristate 13-acetate; Ctrli, initial concentration of the protein in the control culture medium; PMAi, initial concentration of the protein in the culture medium of PMA-treated THP-1 cells.

**Figure 5 ijms-25-09624-f005:**
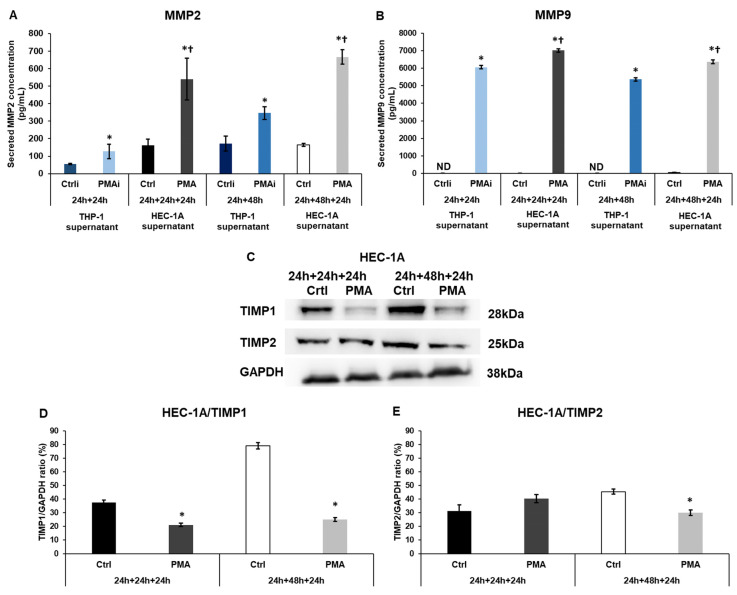
ELISA measurements of MMP2 and MMP9 and the Western blot analyses of TIMP1 and 2 in HEC-1A cells. (**A**) Concentration determination of secreted MMP2 of the non-activated THP-1 cells (Ctrli), PMA-activated THP-1 cells (PMAi), non-activated THP-1 culture medium-treated HEC-1A cells (Ctrl), and PMA-activated conditioned medium-treated HEC-1A cells (PMA). (**B**) Concentration determination of secreted MMP9 of the non-activated THP-1 cells (Ctrli), PMA-activated THP-1 cells (PMAi), non-activated THP-1 culture medium-treated HEC-1A cells (Ctrl), and PMA-activated conditioned medium-treated HEC-1A cells (PMA). (**C**) Western blots of TIMP1 and 2 of HEC-1A cells. (**D**) Analysis of TIMP1 protein level of HEC-1A cells. (**E**) Analysis of TIMP2 protein levels of HEC-1A cells. The samples in the Western blots are: 1. Ctrl 24 h + 24 h + 24 h: medium incubation on non-activated THP-1 cells for 24 h; medium incubation on non-activated THP-1 cells after medium exchange for 24 h; addition of 24 h conditioned medium of non-activated THP-1 cells to HEC-1A cells for 24 h. 2. PMA 24 h + 24 h + 24 h: medium incubation on PMA-activated THP-1 cells for 24 h; incubation on PMA-activated THP-1 cells after medium exchange for 24 h; addition of 24 h conditioned medium of PMA-activated THP-1 cells to HEC-1A cells for 24 h. 3. Ctrl 24 h + 48 h + 24 h: medium incubation on non-activated THP-1 cells 24 h; medium incubation on non-activated THP-1 cells after medium exchange for 48 h; addition of 48 h conditioned medium of non-activated THP-1 cells to HEC-1A cells for 24 h. 4. PMA 24 h + 24 h + 24 h: incubation on PMA-activated THP-1 cells for 24 h; incubation on PMA-activated THP-1 cells after medium exchange for 48 h; addition of 48 h conditioned medium of PMA-activated THP-1 cells to HEC-1A cells for 24 h. Human MMP2 and MMP9 ELISA kits were used to determine secreted MMP2 and MMP9 concentrations according to the manufacturer’s instructions. In the case of WB experiments, the same amount of protein from each sample was separated into SDS-polyacrylamide gels. After blotting, TIMP1 and TIMP2 antibodies were used according to the manufacturer’s protocols. GAPDH was used as the loading control. The blots are representative images. The columns show the mean values ± SD. In (**A**,**B**), the * means *p* < 0.05 compared to Ctrli or Ctrl, respectively, and the † indicates *p* < 0.05 compared to PMAi. In (**D**,**E**), the * shows *p* < 0.05 compared to the control. Abbreviations: MMP2, matrix metalloproteinase 2; MMP9, matrix metalloproteinase 9; PMA, phorbol 12-myristate 13-acetate; Ctrli, initial concentration of the protein in the control culture medium; PMAi, initial concentration of the protein in the culture medium of PMA-treated THP-1 cells, TIMP1 and 2, tissue inhibitors of metalloproteinases; GAPDH, glyceraldehyde 3-phosphate dehydrogenase.

**Figure 6 ijms-25-09624-f006:**
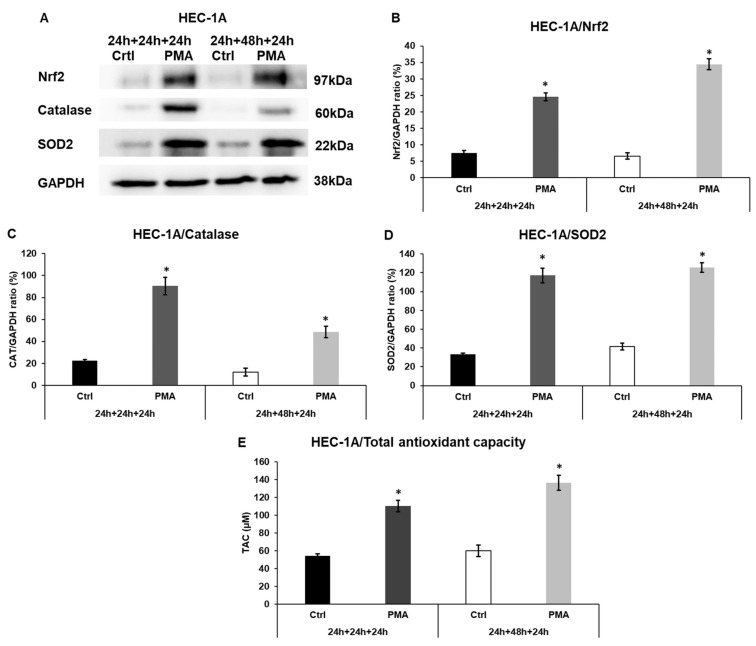
Western blot analyses of Nrf2 transcription factor, catalase, and superoxide dismutase 2 in HEC-1A cells, and the measurement of the total antioxidant capacity in the endometrium cells. (**A**) Western blots of Nrf2, catalase, and SOD2 of HEC-1A cells. (**B**) Analysis of Nrf2 protein level in HEC-1A cells. (**C**) Analysis of catalase protein level in HEC-1A cells. (**D**) Analysis of SOD2 protein levels in HEC-1A cells. (**E**) Total antioxidant capacity of HEC-1A cells. The samples in the Western blot and TAC measurement are: 1. Ctrl 24 h + 24 h + 24 h: medium incubation on non-activated THP-1 cells for 24 h; medium incubation on non-activated THP-1 cells after medium exchange for 24 h; addition of 24 h conditioned medium of non-activated THP-1 cells to HEC-1A cells for 24 h. 2. PMA 24 h + 24 h + 24 h: medium incubation on PMA-activated THP-1 cells for 24 h; incubation on PMA-activated THP-1 cells after medium exchange for 24 h; addition of 24 h conditioned medium of PMA-activated THP-1 cells to HEC-1A cells for 24 h. 3. Ctrl 24 h + 48 h + 24 h: medium incubation on non-activated THP-1 cells 24 h; medium incubation on non-activated THP-1 cells after medium exchange for 48 h; addition of 48 h conditioned medium of non-activated THP-1 cells to HEC-1A cells for 24 h. 4. PMA 24 h + 24 h + 24 h: incubation on PMA-activated THP-1 cells for 24 h; incubation on PMA-activated THP-1 cells after medium exchange for 48 h; addition of 48 h conditioned medium of PMA-activated THP-1 cells to HEC-1A cells for 24 h. For the total antioxidant capacity measurements, the Total Antioxidant Capacity Assay kit was used. TAC was expressed as µM of Trolox equivalent. In the case of WB experiments, the same amount of protein from each sample was separated into SDS-polyacrylamide gels. After blotting, Nrf2, catalase, and SOD2 antibodies were used according to the manufacturer’s protocols. GAPDH was used as the loading control. The blots are representative images. The columns show the mean values ± SD. The * shows *p* < 0.05 compared to the control. Abbreviations: TAC, total antioxidant capacity; CAT, catalase; SOD2, superoxide dismutase 2; PMA, phorbol 12-myristate 13-acetate; GAPDH, glyceraldehyde 3-phosphate dehydrogenase.

**Figure 7 ijms-25-09624-f007:**
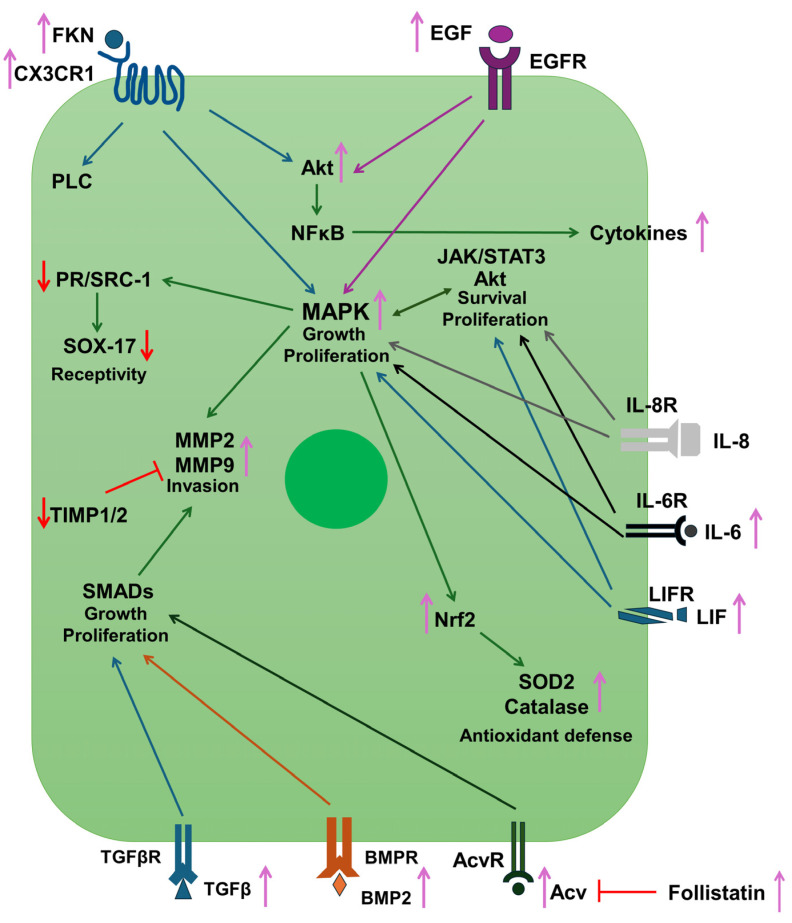
Summary of the results and the proposed mechanisms of action of the examined proteins and signaling pathways in HEC-1A cells. Macrophage-derived factors increase the fractalkine, IL-6, LIF, and TGFβ secretion of HEC-1A endometrial cells, activating the MAPK signaling pathway. Moreover, the treatment of HEC-1A cells with the conditioned medium of the activated THP-1 cells elevated the production of the EGF, which can contribute to MAPK activation and proliferation. The MAPK pathway regulates the MMP2 and MMP9 involved in the invasion, which are also under the control of activin, BMP2, and TGFβ-regulated SMAD transcription factors. The MAPK signaling enhances the synthesis of the Nrf2 antioxidant transcription factor, which in turn activates the production of SOD2 and catalase antioxidant enzymes. The FKN/CX3CR1 interaction supports the expression of the pro-inflammatory cytokines.

**Table 1 ijms-25-09624-t001:** Real-time primer list.

Primer	Sequence 5′ → 3′
Activin forward	TGTTCCAATATGATTCCACCC
Activin reverse	CCACTTGATTTTGGAGGGAT
BMP2 forward	TAAGTTCTATCCCCACGGAG
BMP2 reverse	AGCATCTTGCATCTGTTCTC
CX3CR1 forward	ATTTGTTGGTAGTGTTTGCC
CX3CR1 reverse	CAGGTTCAGGAGGTAAATGT
Cyclin D forward	CTGGTGAACAAGCTCAAGTG
Cyclin D reverse	GCGGATGATCTGTTTGTTCT
Follistatin forward	CAAAGCAAAGTCCTGTGAAG
Follistatin reverse	CCTCTCCCAACCTTGAAATC
Fractalkine forward	TACCTGTAGCTTTGCTCATC
Fractalkine reverse	GTCTCGTCTCCAAGATGATT
GAPDH forward	TGTTCCAATATGATTCCACCC
GAPDH reverse	CCACTTGATTTTGGAGGGAT
Progesterone receptor forward	CCAAAGGCCGCAAATTCT
Progesterone receptor reverse	TGAGGTCAGAAAGGTCATCG
SOX-17 forward	CAGTATCTGCACTTCGTGTG
SOX-17 reverse	AGTAATATACCGCGGAGCTG
SRC-1 forward	AGACCCAACCTTTATTCCCA
SRC-1 reverse	GGTGTTACTTGAACAGGCAT

## Data Availability

The original contributions presented in this study are included in the article; further inquiries can be directed to the corresponding author.

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
