# Peer review of "Activated THP-1 Macrophage-Derived Factors Increase the Cytokine, Fractalkine, and EGF Secretions, the Invasion-Related MMP Production, and Antioxidant Activity of HEC-1A Endometrium Cells"

_ijms, 2024, doi:10.3390/ijms25179624_

Round 1

Reviewer 1 Report

Comments and Suggestions for Authors

The manuscript focuses on soluble factor signaling between macrophages and endometrium, using immortalized cell models (THP1 and HEC1A). Primary endpoints were ELISA and western blotting of key signaling molecules. The methodology needs to be better explained in line with the results and some of the statistically approaches need to be revised (or conclusions adjusted). However, the methods section is appropriately detailed and statistics largely appropriate. Further, in general the conclusions go farther than the presented data and need to be revised. While there is a large amount of interesting data in the manuscript, the writing and presentation must be clarified and narrowed prior to publication.

1) Re-cep-tor in abstract needs to be fixed.

2) In the intro “Activation of the monocytes...” the “the” should be removed. Overall the document should be carefully checked for grammar and typos.

3) While M1/M2 are classic descriptors of macrophage phenotype, more recently the field has shifted to a more nuanced understanding. This and recent references should be included the introduction.

4) The last two paragraphs on the introduction are too strong and need to be dramatically scaled back. Nothing is generally “proven” in a relatively small study using cell lines in vitro.

5) This especially applies to claims regarding “proliferation and growth” which are not measured.

6) The results section is very hard to follow. PMA and PMAi need to be more carefully defined, and the timepoints need to be more carefully explained. The controls are also not clear. Is it conditioned media from unstimulated monocytes?

7) “Based on the results, it can be supposed that the macrophage-derived factors trigger endometrial cell proliferation but provide a deteriorating effect on the receptivity-related proteins.” Proliferation is not measured, and it is unclear what “deteriorating effect” means

8) The order of Figure 2 is off

9) It would be very helpful if the subfigures were clearly labeled as to what they are measuring (eg conditioned media or endometrium response)

10) “suggesting the importance of this chemokine in the regulation of the endometrium.” It’s not clear that that is suggested by this data.

11) “The alterations in the cytokine secretions of the endometrium cells showed a correlation with the time of the incubation of the THP-1 culture medium” The statistical model used does not show a correlation with time. To make that claim you would need to use a statistical model that explicitly included the time dimensions

12) For Supp Fig 1, did you mean TOX8 instead of TOXO8? TOX8 is a resazurin-based assay. While it can be influence by cell proliferation, it is a measure of oxidoreductive capacity and could be unrelated to proliferation and division. The language discussing this should be adjusted. This is especially concerning considering the large shift in antioxidant capacity noted.

13) “Based on these results, it can be hypothesized that the activated macrophages provoke the invasion and may promote the implantation.” This is far too strong.

14) The discussion needs to be revised to reflect the more nuanced understanding of macrophage phenotype.

15) the conclusions should be more limited and narrow considering the limited scope of these in vitro experiments.

16) the limitations of using cell lines is briefly mentioned, but this needs to be discussed more thoroughly.

17) The methods section is very thorough.

Comments on the Quality of English Language

English language is largely fine, but should be edited for clarity.

Author Response

Answers to Reviewer 1.

The manuscript focuses on soluble factor signaling between macrophages and endometrium, using immortalized cell models (THP1 and HEC1A). Primary endpoints were ELISA and western blotting of key signaling molecules. The methodology needs to be better explained in line with the results and some of the statistically approaches need to be revised (or conclusions adjusted). However, the methods section is appropriately detailed and statistics largely appropriate. Further, in general the conclusions go farther than the presented data and need to be revised. While there is a large amount of interesting data in the manuscript, the writing and presentation must be clarified and narrowed prior to publication.

1) Re-cep-tor in abstract needs to be fixed.

Thank you for the comment. The word has been fixed.

2) In the intro “Activation of the monocytes...” the “the” should be removed. Overall the document should be carefully checked for grammar and typos.

Thank you for your comment. The whole manuscript has been checked and corrected.

3) While M1/M2 are classic descriptors of macrophage phenotype, more recently the field has shifted to a more nuanced understanding. This and recent references should be included the introduction.

Thank you for the advice. The introduction has been supplemented with additional classifications and references. Please see lines 55-69.

4) The last two paragraphs on the introduction are too strong and need to be dramatically scaled back. Nothing is generally “proven” in a relatively small study using cell lines in vitro.

5) This especially applies to claims regarding “proliferation and growth” which are not measured.

Thank you for the comments 4-5. The last two paragraphs of the introduction were modified. Please see lines 89-109.

6) The results section is very hard to follow. PMA and PMAi need to be more carefully defined, and the timepoints need to be more carefully explained. The controls are also not clear. Is it conditioned media from unstimulated monocytes?

In the case of ELISA measurements, the figure legends have been supplemented with the following explanation: Concentration determination of secreted {name of the protein} of the non-activated THP-1 cells (Ctrli), PMA-activated THP-1 cells (PMAi), non-activated THP-1 culture medium-treated HEC-1A cells (Ctrl), and PMA-activated conditioned medium treated HEC-1A cells (PMA).

In the case of WBs, the figure legends have been strengthened with additional sample information: “The samples in the Western blots, viability, and ATP measurements are:

  1. Ctrl 24h+24h+24h: medium incubation on non-activated THP-1 cells for 24h; medium incubation on non-activated THP-1 cells after medium exchange for 24h; 24h conditioned medium of non-activated THP-1 cells added to HEC-1A cells for 24h.
  2. PMA 24h+24h+24h: medium incubation on PMA-activated THP-1 cells for 24h; incubation on PMA-activated THP-1 cells after medium exchange for 24h; 24h conditioned medium of PMA-activated THP-1 cells added to HEC-1A cells for 24h.
  3. Ctrl 24h+48h+24h: medium incubation on non-activated THP-1 cells 24h; medium incubation on non-activated THP-1 cells after medium exchange for 48h; 48h conditioned medium of non-activated THP-1 cells added to HEC-1A cells for 24h.
  4. PMA 24h+24h+24h: incubation on PMA-activated THP-1 cells for 24h; incubation on PMA-activated THP-1 cells after medium exchange for 48h; 48h conditioned medium of PMA-activated THP-1 cells added to HEC-1A cells for 24h.

The materials and methods section 2.1. describes the sample types and treatments.

7) “Based on the results, it can be supposed that the macrophage-derived factors trigger endometrial cell proliferation but provide a deteriorating effect on the receptivity-related proteins.” Proliferation is not measured, and it is unclear what “deteriorating effect” means

We apologize for the inappropriate wording.

The macrophage-derived factors have a decreasing effect on the expression of receptivity-related proteins. The questionable part of the manuscript was corrected. “Based on the results, it can be supposed that the macrophage-derived factors act on the activin receptor and BMPR signaling pathways by modifying the ligand expression, which can influence the cell cycle of the endometrial cells. On the other hand, they decrease the levels of the receptivity-related proteins.” Please see lines 104-108.

We measured the ATP content of HEC-1A cells after the treatments, supplementing the results of the cell viability measurements. ATP measurement is a good marker of cell proliferation since a higher ATP level indicates a higher living cell number. According to the increased phosphorylation of ERK1/2 supposing the activation of the signaling pathway, real-time PCRs were evaluated on the cyclin D gene of HEC-1A cells, which showed significant elevation in the mRNA expression, suggesting the activation of the cell cycle.

The results are introduced in the manuscript in Figure 4.

8) The order of Figure 2 is off

Thank you for the comment. The order of the subfigures, as well as the description in the text, have been corrected.

9) It would be very helpful if the subfigures were clearly labeled as to what they are measuring (eg conditioned media or endometrium response)

Thank you for the advice. All figures have been modified, and the figure legends have been supplemented with the exact sample names and treatment setups.

10) “suggesting the importance of this chemokine in the regulation of the endometrium.” It’s not clear that that is suggested by this data.

Thank you for the comment. The questionable sentence has been corrected. Please see lines 143-146.

11) “The alterations in the cytokine secretions of the endometrium cells showed a correlation with the time of the incubation of the THP-1 culture medium” The statistical model used does not show a correlation with time. To make that claim you would need to use a statistical model that explicitly included the time dimensions

We apologize for the wrong phrasing. We did not apply statistical analysis for time dimensions. The sentence has been revised. Please see lines 177-179.

12) For Supp Fig 1, did you mean TOX8 instead of TOXO8? TOX8 is a resazurin-based assay. While it can be influence by cell proliferation, it is a measure of oxidoreductive capacity and could be unrelated to proliferation and division. The language discussing this should be adjusted. This is especially concerning considering the large shift in antioxidant capacity noted.

Thank you for the question. The name of the kit is TOX8, and it was used to analyze the viability of HEC-1A cells. We also measured the ATP levels of HEC-1A cells, supporting our hypothesis about the increasing cell number. According to the increased phosphorylation of ERK1/2 supposing the activation of the signaling pathway, real-time PCRs were evaluated on the cyclin D gene of HEC-1A cells, which showed significant elevation in the mRNA expression, suggesting the activation of the cell cycle. The results can be found in Figure 4.

13) “Based on these results, it can be hypothesized that the activated macrophages provoke the invasion and may promote the implantation.” This is far too strong.

The sentence was corrected. Please see lines 351-352.

14) The discussion needs to be revised to reflect the more nuanced understanding of macrophage phenotype.

Thank you for the advice. The discussion has been revised and completed. Please see lines 594-620.

15) the conclusions should be more limited and narrow considering the limited scope of these in vitro experiments.

Thank you for the advice. The conclusion has been revised and rewritten. Please see lines 594-606.

16) the limitations of using cell lines is briefly mentioned, but this needs to be discussed more thoroughly.

Thank you for the comments. The limitations have been discussed more on page 17, lines 747-754.

17) The methods section is very thorough.

Thank you for the comment. According to the advice of the second reviewer, the methods have been supplemented with additional descriptions and references.

Comments on the Quality of English Language

English language is largely fine, but should be edited for clarity.

Thank you. The manuscript has been edited.

All changes can be followed by the track change mode.

Reviewer 2 Report

Comments and Suggestions for Authors

Pandur et al. present a manuscript that contains interesting and well-structured results. The authors provide an adequate introduction to the state of the art that justifies the novelty of the study. The results are correct, with an adequate discussion, which means that the conclusions are correctly supported. The methodology is conservative but adequate for the stated objectives. However, the authors should address several points:

-First, the title is too unspecific. The authors should improve the title, being precise.

-The figures are of low quality. The authors should unify formats and improve the quality extensively. The authors should use color in the figures.

-The figures should be more self-explanatory. The figure legends should be improved.

-The methodology should be better described. Please provide specific references.

-Table 1 should contain more information. Temperature? Number of cycles?

-The authors should make a graphic summary of the results, where the proposed mechanisms are clear.

-Authors must improve grammatical aspects in an extensive way.

Comments on the Quality of English Language

The English is very difficult to understand/incomprehensible.

Author Response

Answers to Reviewer 2.

Pandur et al. present a manuscript that contains interesting and well-structured results. The authors provide an adequate introduction to the state of the art that justifies the novelty of the study. The results are correct, with an adequate discussion, which means that the conclusions are correctly supported. The methodology is conservative but adequate for the stated objectives. However, the authors should address several points:

-First, the title is too unspecific. The authors should improve the title, being precise.

Thank you for the advice. The title of the manuscript has been changed to “Activated THP-1 macrophages-derived factors increase the cytokines, fractalkine, and EGF secretions, the invasion-related MMP production and antioxidant activity of HEC-1A endometrium cells”.

-The figures are of low quality. The authors should unify formats and improve the quality extensively. The authors should use color in the figures.

All figures have been colored and have been exchanged to unify and improve the quality. Some WBs have also been exchanged to increase the quality. The figures have 600 dpi in resolution, which is in line with the expectation of the journal.

-The figures should be more self-explanatory. The figure legends should be improved.

Thank you for the advice. The figures and legends have been modified for better understanding.

-The methodology should be better described. Please provide specific references.

Thank you for the comments. The methods were supplemented with more information and references.

-Table 1 should contain more information. Temperature? Number of cycles?

The requested data have been included in the section 4.2. Please see lines 665-677.

-The authors should make a graphic summary of the results, where the proposed mechanisms are clear.

Thank you for the advice. A new figure describing the results and summarizing the proposed mechanisms has been created. Please see Figure 7. on page 12.

-Authors must improve grammatical aspects in an extensive way.

Thank you for the comment. The manuscript has been revised.

Comments on the Quality of English Language

The English is very difficult to understand/incomprehensible.

Thank you for the comment. The manuscript has been revised.

All changes can be followed by the track change mode.

Round 2

Reviewer 1 Report

Comments and Suggestions for Authors

Concerns have been addressed.

Reviewer 2 Report

Comments and Suggestions for Authors

Accept in present form.